# Predictive Factors of Athyroglobulinemia After Total Thyroidectomy for Papillary Thyroid Cancer

**DOI:** 10.3390/cancers16244129

**Published:** 2024-12-11

**Authors:** Marta Fernández-Baeza, Nuria V. Muñoz-Pérez, Ignacio Roldán-Ortiz, María J. Alonso-Sebastián, Francisco M. Carbajo-Barbosa, Rafael Rejón-López, María C. Olvera-Porcel, Antonio Becerra-Massare, Juan I. Arcelus-Martínez, Jesús María Villar-del-Moral

**Affiliations:** 1Endocrine Surgery Unit, General Surgery Department, Hospital Universitario Virgen de las Nieves, 18014 Granada, Spain; nuriav.munoz.sspa@juntadeandalucia.es (N.V.M.-P.); mjose.alonso.sspa@juntadeandalucia.es (M.J.A.-S.); franciscomanuel.carbajobarbosa@gmail.com (F.M.C.-B.); rafael.rejon.sspa@juntadeandalucia.es (R.R.-L.); jesusm.villar.sspa@juntadeandalucia.es (J.M.V.-d.-M.); 2Medical School, University of Granada, 18071 Granada, Spain; 3Instituto de Investigación Biosanitaria ibs.GRANADA, 18012 Granada, Spain; 4Biostatistics, Unidad de Gestión y Apoyo a la Investigación, Hospital Universitario Virgen de las Nieves, 18014 Granada, Spain

**Keywords:** athyroglobulinemia, thyroglobulin, papillary thyroid cancer, total thyroidectomy, prognosis

## Abstract

Thyroglobulin is the specific tumor marker for papillary and other types of epithelial thyroid cancer. It holds significant value after total thyroidectomy, as elevated or rising levels indicate tumor persistence or recurrence. Therefore, from a biochemical standpoint, the goal of surgery in papillary thyroid cancer is to achieve undetectable levels of postoperative thyroglobulin. In this unicentric retrospective study, basal non-stimulated postoperative athyroglobulinemia was obtained in 89.6% of a series of 202 patients. We have found that the achievement of undetectable levels was more closely related to factors that have to do with tumor stage (tumor diameter, lymph node spread, and metastatic disease) than with differences in epidemiological data, clinic manifestations, preoperative diagnosis, and some histological features such as multifocality or the presence of aggressive cytological variants.

## 1. Introduction

Papillary thyroid cancer (PTC) is the most common malignant thyroid neoplasm [1], approximately accounting for 85–90% of epithelial thyroid carcinomas [2], with a worldwide increasing incidence [3,4]. The standard treatment is surgery by hemithyroidectomy or total thyroidectomy (TT) [1], with or without lymph node dissection, followed in high-risk cases by adjuvant therapies such as radioactive iodine (I^131^) ablation [5,6]. Although it generally has a favorable prognosis, with a low mortality rate [6] and high cure rates, certain patient subgroups are at higher risk of recurrence or metastasis. Consequently, the main objective in its follow-up is the early detection of persistent or recurrent disease [7]. This, in addition to the increasing number of patients treated with hemithyroidectomy following the publication of the ATA 2015 guidelines [6,8,9], underscores the need to identify reliable prognostic factors to ensure optimal management of this disease.

The primary biomarker used in PTC monitoring is thyroglobulin (Tg) [1], a glycoprotein produced exclusively by normal and tumoral thyroid epithelium [2]. Although its preoperative detection is not useful in the differential diagnosis of thyroid conditions [10], it has a significant postoperative role. It is considered an ideal tool for monitoring PTC patients surgically treated through TT, enabling the detection of persistent or recurrent disease when its levels increase [2,10,11].

Therefore, the surgical goal can be assumed to be achieving biochemical remission [10], that is, the absence of dTg in peripheral blood, a phenomenon known as athyroglobulinemia (uTg). However, uTg has limitations, as its levels can be influenced by other factors, including thyroid tissue destruction after surgery or I^131^ ablation [6], the production of antithyroglobulin antibodies (TgAb) [2], or tumor differentiation. Thus, identifying predictive factors that promote or hinder uTg achievement would be essential to tailor PTC patient follow-up, provide accurate information, and optimize therapeutic strategies.

This study aims to identify which demographic, clinical, histopathological, and therapeutic variables influence the achievement of basal postoperative uTg in patients undergoing TT for PTC in our context.

## 2. Materials and Methods

### 2.1. Study Design and Patient Characteristics

A retrospective single-center study was conducted at a regional hospital affiliated with the European Registry of Endocrine Surgery, Eurocrine®. Its aim was to evaluate basal postoperative Tg levels in patients with PTC. All cases of TT (initial or after completion thyroidectomy) with a histopathological diagnosis of PTC, registered in Eurocrine®, in an eight-year period from January 2015 to April 2023, were included.

### 2.2. Exclusion Criteria

Patients in whom postoperative Tg determination had not been performed were excluded. Also excluded were cases with postsurgical TgAb levels above 4.1 IU/mL, the upper limit of normal range in our center, as these antibodies may interfere with Tg measurement, resulting in false negative cases [12,13].

### 2.3. Analyzed Variables

They were obtained from the Eurocrine® registry and the electronic medical records of the patients. Several data were collected and can be grouped into the following:Demographic variables

Gender and age at the time of surgery.

2.Clinical variables

Presence or not of intrathoracic growth of the gland, main indication for surgery (segmented in benign pathology, exclusion of malignancy, or suspected/confirmed malignancy). It was also analyzed whether the intervention was initial surgery or to complete a previous hemithyroidectomy.

3.Cytological results

Based on the Bethesda classification for reporting thyroid nodule cytology [8,14].

4.Therapeutic variables
-Extent of surgical resection: TT (with or without removal of prethyroid muscles) versus additional resection of adjacent structures (esophageal wall, larynx, trachea, recurrent nerve, internal jugular vein, etc.).-Performance of a neck dissection or not.-Indication for surgery: suspicion of malignancy or malignancy, thyrotoxicosis, and compressive symptoms.5.Histopathological variables
-Size of the primary tumor.-Tumor multifocality, defined as the presence of more than one tumor focus in the thyroid gland.-Type of papillary cancer, distinguishing between classic variety, encapsulated variant, follicular variant, or aggressive variants (hobnail, tall cell, columnar cell, cribriform/morular, or diffuse sclerosing PTC).-Classification according to the TNM system.-Presence or absence of lymphocytic thyroiditis in the thyroidectomy specimen.6.Biochemical variables
-Basal Tg: serum Tg levels were measured under basal conditions, without suppression therapy or administration of recombinant thyrotropin (TSH), with levels equal to or less than 1 ng/mL [15] considered indicative of uTg and levels above this threshold classified as dTg. This was considered the dependent variable of the study.-TgAb: measured concurrently with Tg determination.7.Estimation of recurrence and disease-specific mortality risk

According to MACIS score (metastasis, age, completeness of resection, invasion, and size) [16].

For its calculation, the following formula was used:

MACIS = 3.1 × age (if age > 39 years) + 0.3 × tumor size (in cm) + 1 (distant metastasis) + 1 (extracapsular invasion) + 0.5 (incomplete resection).

Based on the result, patient’s risk was classified as high (greater than 8 points), intermediate (between 6 and 8 points), or low (less than 6 points).

8.Time of follow-up

### 2.4. Statistical Analysis

A descriptive statistical analysis was performed where quantitative variables reported in this manuscript were expressed as median and interquartile range (i.q.r.) due to the lack of normality in the distribution of the data, verified by the Kolmogorov–Smirnov test unless otherwise specified. Categorical variables are reported as counts and percentages. Subsequently, a Mann–Whitney U test was used to compare the results of quantitative variables among groups (patients with uTg or not). Categorical variables were compared with χ^2^ or Fisher’s exact test. The association of each independent variable with the likelihood of having postoperative uTg was assessed by calculating the corresponding crude odds ratios (cORs).

All variables with *p* ≤ 0.05 in the bivariate analysis were considered candidates for inclusion in a potential multivariate model by calculating adjusted odds ratios (aORs). Variables were excluded from the final multivariate logistic regression model based on the results of the likelihood ratio test. Additionally, 95% confidence intervals (CI) were calculated for OR. The significance level was stated at 0.05. The software used for the calculations was STATA vs. 16.

## 3. Results

### 3.1. Demographic, Clinical, Cytological, and Surgical Characteristics

In the period previously indicated, 286 patients underwent TT due to PTC in our institution. As can be seen in the Study Flow Diagram in Figure 1, and after applying the exclusion criteria, the group of patients finally evaluable was made up of 202 cases. Of them, 181 (89.6%) reached postoperative uTg, whereas in 21 remained detectable (10.4%).

As shown in Table 1, 71.3% were women, with no differences in the rate of athyroglobulinemia among genders. Regarding age, the median in the series was 53 years (IQR 35–73), with no differences between groups. In terms of the presence of an intrathoracic thyroid component, this was found in 5.5% of patients in the series, with no impact on the achievement of postoperative uTg.

Concerning the main indication for surgery, in 44 cases (21.8%), there was no preoperative suspicion of neoplasm. These patients underwent surgery due to compressive symptoms in 25 cases and thyrotoxicosis in 18. In any case, the preoperative diagnosis did not affect the achievement of postoperative uTg (*p* = 0.132, Fisher’s exact test), although the small number of patients with dTg in cases without preoperative suspicion prevented the determination of cOR and CI.

Similarly, as shown in Table 2, no differences in uTg rates were observed following TT performed in two procedures (required in 14 patients, 6.9% of the series) compared to single-operation thyroid resection (*p* = 0.370, Fisher’s exact test). Again, it was not feasible to determine cOR and CI (no patients undergoing the two-step procedure had dTg).

As for the preoperative cytological report, it was available and conclusive in 181 cases (89.6%). It suggested benign condition in 21 (11.6%), indeterminate results (Bethesda III and IV) in 37 (20.4%), and suspicious for malignancy or malignancy (Bethesda V and VI) in 123 (68.0%). Among the 181 patients, 20 (11.0%) had dTg, with no differences among groups found according to cytological results (*p* = 0.212, Fisher’s exact test). For this variable, the determination of cOR and CI was also not feasible.

Regarding the extent of resection, isolated TT, with or without prethyroid muscle resection, was performed in 183 patients (90.6%). In 19 cases (9.4%), en-bloc resection of one or more adjacent structures, such as the recurrent laryngeal nerve, trachea, larynx, esophageal wall, internal jugular vein, or carotid artery, was required. Extended resection increased the likelihood of not achieving uTg (cOR 5.169) fivefold compared to patients who underwent standard resection, with statistically significant differences. This means that patients who underwent extended resection were 5.169 times more likely not to present uTg than to present it, compared to those who underwent non-extended resection.

In 73 patients (36.4%), a neck dissection was added to the thyroidectomy: in 64 cases, a central neck dissection and, in nine, both central and lateral compartments. This situation increased the risk of maintaining detectable Tg more than twice (cOR 2.6) compared to patients who did not undergo neck dissection, with statistically significant differences in logistic regression.

### 3.2. Histopathological Characteristics and Tumor Staging Features

These are summarized in Table 3 and Table 4.

As shown in Table 3, multicentricity was observed in 25.7% of cases. Its presence did not impact the risk of maintaining dTg after surgery. Nor did the histological variant of papillary carcinoma (*p* = 0.641, Mann–Whitney U test), although logistic regression analysis was not feasible due to the small number of patients with dTg in certain subgroups. In 39 patients (19.3%), histological signs of lymphocytic thyroiditis were found in the thyroidectomy specimen; however, this finding had no impact on the achievement of uTg.

As summarized in Table 4, regarding tumor size, for patients who achieved uTg, the median (iqr) size was 12 (3–21) mm. For those with dTg, it was 20 mm (8–32), with statistically significant differences (*p* = 0.003). Comparing different T-stages, the risk of dTg increases with stage progression, being 6-fold higher for T3b tumors and 27 times higher for T4 patients compared to T1 cases.

In relation to the nodal stage, in 146 cases (72.3%), there was no evidence of lymph node involvement, either because no lymph nodes were detected in the specimen (Nx) in 84 cases (41.6% of the series) or because in 62 patients (30.7%), the biopsied or dissected lymph nodes ruled out metastases (N0). In any case, the presence or absence of lymph node involvement did not generate significant differences in the postoperative uTg rate.

Overall, 10 out of our 202 patients (4.9%) showed distant metastases at disease onset. Naturally, the presence of metastatic spread was a strong predictor of postoperative uTg. Regarding the MACIS score, 119 patients (58.9%) had a low score, below six points. For 67 (33.2%), the obtained rating was considered intermediate (six to eight points). In 16 cases (7.9%), the scoring was consistent with high risk (>8 points). Low or intermediate scores were more frequently associated with achieving basal uTg compared to patients with high MACIS scores (*p* < 0.000).

Due to the limited number of variables with statistical significance in the bivariate logistic regression study, multivariate analysis could not be performed.

### 3.3. Follow-Up

The median follow-up for these patients was 34 months (IQR 12-72).

## 4. Discussion

Papillary thyroid cancer, despite being a neoplasm with a generally favorable prognosis [6], poses significant challenges in postoperative follow-up, mostly in high-risk cases. One of the main challenges in managing these patients is the correct interpretation of Tg levels, a molecule that has become the primary tumor marker in the follow-up of PTC cases after TT, as postoperative uTg indicates effective disease control. In our series, a high percentage of patients (89.6%) achieved basal uTg, underscoring the effectiveness of the surgical approach applied. This finding is particularly relevant since the factors influencing the achievement of uTg in this patient group were primarily related to tumor extension. Among them were tumor size, adjacent structure involvement, distant metastatic involvement, distant metastases, and, consequently, higher MACIS scores. These data suggest that the extent of the tumor at diagnosis and precise surgical resection are determinants of success in the biochemical follow-up of PTC.

When compared with other studies, as seen in the work by Tuttle et al. [17], variable Tg levels (the study by Robbins et al. [18] reported between 0 and 7.6 ng/mL) are observed depending on factors such as tumor size and the extent of surgical resection. The variability of Tg according to the extent of resection is also evident in the study by Lang et al. [19], which evaluated Tg values depending on whether unilateral prophylactic lymphadenectomy was performed, reporting Tg values between <0.5 and 6.7 ng/mL depending on the group, although this study evaluated TSH-stimulated Tg. The consistency between these results and previously published studies supports the hypothesis that factors related to tumor staging, rather than demographic or clinical patient characteristics, play a determining role in achieving postoperative uTg.

In our study, demographic variables such as age and sex did not significantly impact the development of uTg. Although previous studies have suggested that male sex and age >45 years are associated with a worse prognosis in PTC [17], these variables do not appear to directly influence the achievement of uTg.

Intrathoracic growth of the thyroid gland did not show a significant association with the achievement of postoperative uTg in our patients. Previous studies, such as that published by Ríos et al. [20], highlight that surgery for intrathoracic goiter is more related to relieving the symptoms caused by the goiter than to the risk of disease recurrence, as the intrathoracic extension does not correlate with a worse prognosis.

Examining the main indication for surgery, although it would be reasonable to assume that malignancy or suspected malignancy as surgical indication could be associated with a lower uTg rate compared to benign indications, no differences among groups regarding uTg rate were found. This could be attributed to the small number of patients with dTg in the benign indication group. No studies have addressed this topic, making it a potentially interesting area for future research.

Regarding the performance of thyroidectomy as a one or two-step procedure, no difference was found concerning postoperative uTg rate. This issue represents a topic to be addressed in future research, as no studies have been published on this subject.

In analyzing histology, preoperative cytology, evaluated through the Bethesda classification, showed no relevant relationship with uTg achievement in this study. In our series, patients with benign or indeterminate cytological results (Bethesda III and IV) and those with suspected or confirmed malignancy (Bethesda V and VI) achieved similar uTg rates. The report from Cibas and Ali [14] emphasized that the Bethesda cytological stratification is useful for diagnosis and preoperative planning, but concerning biochemical disease follow-up, the results of this study align with those described by the American Association of Clinical Endocrinologists and the American Association of Endocrinologists, which established that postoperative biochemical predictive factors appear to be more closely related to tumor extent than to preoperative cytology [21].

In our series, extended surgical resection including adjacent structures was significantly associated with a lower likelihood of achieving uTg. In this way, more extensive tumor invasion implies greater options to maintain microscopic disease and accordingly maintain detectable Tg. Previous studies also highlighted this relationship, linking the invasion of adjacent structures, including lymph node involvement [4], with persistent Tg after PTC surgery.

Lymphadenectomy was associated with a higher likelihood of maintaining detectable postoperative Tg. This result aligns with previous research, such as that by Hay et al. [22]. They noted that the presence of lymph node metastases indicates a more advanced disease and, therefore, a lower likelihood of achieving uTg. Consequently, the addition of lymphadenectomy, especially when performed with therapeutic intent, appears to reflect a higher-risk profile, suggesting that these patients may benefit from more thorough follow-up and adjuvant treatments.

The relationship between tumor size (T) and the likelihood of achieving uTg is one of the most consistent findings of this study. Patients with a lower T stage achieved uTg more frequently than those with a higher staging. This finding aligns with the existing literature, which reports that larger tumors are more likely to spread to lymph nodes and therefore have a poorer prognosis, associated with a lower success rate in achieving uTg [4]. This relationship highlights the importance of early detection and adequate surgical control in the initial stages of PTC, as this approach can improve postoperative biochemical outcomes and reduce the risk of tumor persistence.

In our study, the presence of multicentricity was not significantly associated with achieving uTg. Multicentricity has been described as a common feature in PTC [23], occurring in up to 50% of papillary cancer cases, with a negative impact on prognosis in some reports [23,24], as its presence appears to suggest a higher potential for metastatic spread, tumor persistence, and an increased risk of recurrence [24]. Given this assumption, it can be expected that multicentrity would be associated with a lower percentage of uTg achievement, a relationship not demonstrated in this study and not analyzed in previous reports.

In this series, the histological variant of PTC also showed no significant impact on achieving postoperative uTg, although histological differentiation may be a relevant predictor in terms of survival and tumor recurrence due to the higher risk of metastatic disease in more aggressive variants [25], the influence of the histological subtype on postoperative biochemistry could be mitigated by complete surgical resection and the absence of additional poor prognostic factors. However, further studies are needed on this topic.

Additionally, lymphocytic thyroiditis did not show a significant relationship with uTg achievement in our study. Some studies have indicated that lymphocytic thyroiditis may be associated with a favorable prognosis in PTC, as lymphocytic infiltration may act as an antitumor immune response [26]. However, this study’s results are consistent with previous studies suggesting that, although lymphocytic thyroiditis does not influence the risk of tumor recurrence [27]. Authors such as Latrofa et al. [28] indicated that lymphocytic thyroiditis is often associated with TgAb detection and therefore a low or undetectable Tg, but it was not necessarily associated with a lower recurrence probability.

Regarding lymph node involvement, we did not observe differences in the postoperative uTg rate according to the lymph node stage. Previous studies have identified lymph node involvement as an important risk factor for tumor recurrence, especially in cases with extensive lymph node dissemination [29,30]. The absence of differences between different levels of lymph node involvement in postoperative Tg status in our study may be due to the small N1b subgroup size, which limits the statistical power of this analysis. The presence of distant metastases was significantly associated with detectable postoperative Tg. The obvious relationship among metastatic spread, biochemical persistence, and poor prognosis is consistent with the available literature [31]. This pattern suggests that patients with distant metastases require intensive follow-up and possibly adjuvant interventions to optimize postoperative Tg and disease control.

Finally, our results support the usefulness of scales like the MACIS score in stratifying the risk of recurrence and mortality in PTC and in surrogate markers as thyroglobulin values. This result is consistent with other studies where the MACIS score is considered a robust predictor of recurrence and tumor persistence, as it integrates key clinical and pathological factors such as metastasis, age, extent of resection, level of invasion, and tumor size [32,33]. These data reinforce the usefulness of this score as a risk stratification tool in the postoperative follow-up of PTC, facilitating the identification of patients who could benefit from more aggressive follow-up strategies and adjuvant therapies.

Regarding this study’s generalizability, to increase its external validity and improve its applicability to broader populations, a multicenter approach could be taken in future studies, allowing for the inclusion of patients from diverse geographic and clinical settings. This would help minimize potential biases arising from the unicentric nature of this study. It would also be interesting to evaluate Tg levels both in basal and stimulated states, assessing their impact on long-term prognosis and recurrence prediction.

Additionally, long-term follow-up of patients could provide data on the usefulness of uTg as a predictive marker of the absence of recurrence. The integration of additional variables, such as genetic or molecular markers, would also contribute to a better understanding and generalization of current findings.

Future research could be focused on assessing the impact of complementary therapies, such as I^131^, on achieving uTg, as well as developing predictive models that integrate multiple clinical-pathological variables. Additionally, prospective studies examining the role of more conservative surgical techniques (hemithyroidectomy) in managing early-stage PTC and the real value of postoperative thyroglobulin in these situations would be valuable.

The main limitation of this study is the small number of patients with detectable Tg (dTg), which impacted the statistical accuracy of the analysis, preventing the use of multivariate analysis. Additionally, other limitations include its retrospective and single-center design, which may limit the generalizability of the results to other populations or centers.

The strengths of this study include the comprehensive analysis of multiple relevant variables with a low rate of missing values in most of them. Furthermore, the series was treated uniformly in terms of diagnostic procedures and therapeutic strategies. The decisions were adopted in all cases under the guidance of a multidisciplinary committee. Lastly, we highlight that the surgical treatment was performed by expert endocrine surgeons with a high volume of interventions.

This article provides a novel perspective by comprehensively analyzing the various factors influencing the achievement of uTg in patients with PTC. While Tg is a well-established marker in the follow-up of PTC, this study takes an innovative approach by specifically examining the factors that influence the achievement of uTg following total thyroidectomy. This contributes to clinical practice by positioning uTg as a key indicator of both surgical success and disease control and allows for the refinement of risk stratification in patients with PTC, helping to identify those who may benefit from more intensive follow-up or adjuvant therapies.

The novelty of the study also lies in its immediate clinical applicability. The article suggests that surgical decisions, such as avoiding unnecessary extensive resections, can optimize uTg rates, thus enabling more personalized therapeutic and follow-up strategies. Additionally, it distinguishes itself from other studies by providing valuable insights that enhance the understanding of the mechanisms and factors involved in tumor progression and recurrence in PTC. This, in turn, could facilitate the optimization of therapeutic strategies and monitoring protocols for these patients.

## 5. Conclusions

In this study, postoperative basal uTg was achieved in nine out of ten PTC cases undergoing total thyroidectomy. Factors associated with tumor extension, such as the need for extensive resections or lymphadenectomies, more advanced T stages, the presence of distant metastases, and high MACIS scores, were predictors of difficulty in achieving postoperative uTg. These findings emphasize the importance of an early and accurate diagnosis, as well as a multidisciplinary approach in the management of these patients.

## Figures and Tables

**Figure 1 cancers-16-04129-f001:**
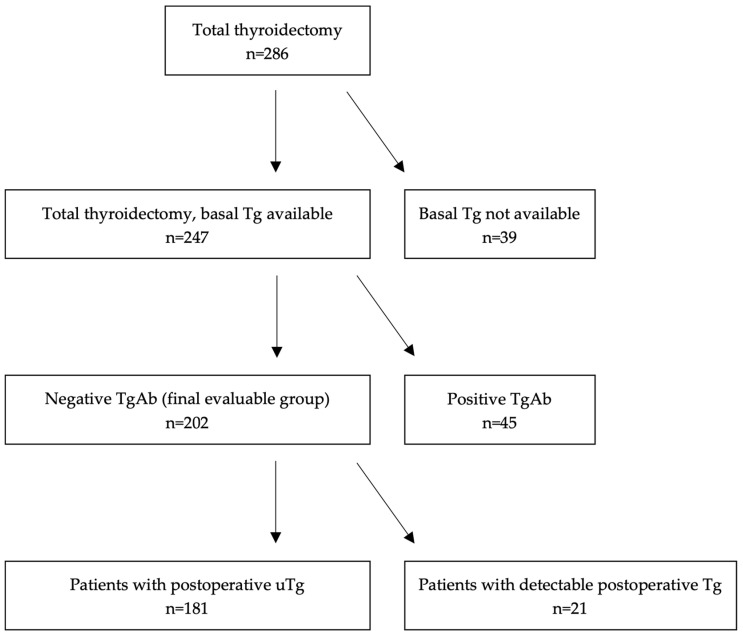
Study flow chart. Tg: thyroglobulin. TgAb: antithyroglobulin antibodies. uTg: indetectable thyroglobulin.

**Table 1 cancers-16-04129-t001:** Demographic and clinical features of 202 patients undergoing total thyroidectomy due to papillary thyroid cancer, 2015–2023. Bivariate analysis among patients with undetectable or detectable basal postoperative thyroglobulin.

				95% CI for cOR		
	uTg 181 (89.6)	dTg 21 (10.4)	cOR	Inferior	Superior	*p* Value
**Sex**						
Female	131 (91.0)	13 (9.0)	Reference category			
Male	50 (86.2)	8 (13.8)	1.612	0.63	4.12	0.319
**Age (years) ***	60 (38/82)	52 (34/70)	1.015	0.99	1.07	0.332
**I** **ntrathoracic goitre**						
Absent	171 (89.5)	20 (10.5)	Reference category			
Present	10 (90.9)	1 (9.1)	1.169	0.14	9.16	0.884
**Main indication for surgery:**						
Compressive symptoms	25 (100)	0 (0.0)	NPC			
Thyreotoxicosis	18 (94.7)	1 (5.3)	NPC			
Excluding malignancy/malignancy	138 (87.3)	20 (12.7)	2.608	0.33	20.62	0.132 ^&^

Values are n (%) unless otherwise indicated. * Median (i.q.r.). CI: confidence interval. cOR: crude odd ratio. uTg: undetectable thyroglobulin. dTg: detectable thyroglobulin. NPC: no possible calculation of cOR and CI due to small number of patients in dTg group. &: Fisher’s exact test. Missing data: sex 0 percent, age 0 percent, intrathoracic goitre 0 percent, and main indication for surgery 0 percent.

**Table 2 cancers-16-04129-t002:** Demographic, clinical, cytologic, and surgical features of 202 patients undergoing total thyroidectomy due to papillary thyroid cancer, 2015–2023. Bivariate analysis among patients with undetectable or detectable basal postoperative thyroglobulin.

				95% CI for cOR		
	uTg 181 (89.6)	dTg 21 (10.4)	cOR	Inferior	Superior	*p* Value
**Performance of thyroidectomy**						
One-step procedure	167 (88.8)	21 (11.2)	NPC			
Two-steps procedure	14 (100.0)	0 (0.0)	NPC			0.370 ^&^
**Report of FNAB cytology**						
Benign	21 (100.0)	0 (0.0)	NPC			
Bethesda III and IV	32 (86.5)	5 (13.5)	NPC			
Bethesda V and VI	108 (87.8)	15 (12.2)	1.125	0.37	3.33	0.212 ^&^
**Extent of local resection**						
Thyroid +/− strap muscles	168 (91.8)	15 (8.2)	Reference category			
Including neighboring structures	13 (68.4)	6 (31.6)	5.169	1.17	15.56	**0.003**
**Concomitant lymph node procedure**						
No or only diagnostic biopsy	120 (93.0)	9 (7.0)	Reference category			
Associated neck dissection	61 (83.5)	12 (16.4)	2.622	1.04	6.56	**0.039**

Values are n (%) unless otherwise indicated. CI: confidence interval. cOR: crude odd ratio. uTg: undetectable thyroglobulin. dTg: detectable thyroglobulin. NPC: no possible calculation of cOR and CI due to small number of patients in dTg group. &: Fisher’s exact test. FNAB: Fine-needle aspiration biopsy. Missing data: performance of thyroidectomy 0 percent, report of FNAB cytology 10.4 percent, extension of local resection 0 percent, and additional lymph node procedure 0 percent.

**Table 3 cancers-16-04129-t003:** Histological features of 202 patients undergoing total thyroidectomy due to papillary thyroid cancer, 2015–2023. Bivariate analysis among patients with undetectable or detectable basal postoperative thyroglobulin.

				95% CI for cOR		
	uTg 181 (89.6)	dTg 21 (10.4)	cOR	Inferior	Superior	*p* Value
**Multicentricity**						
No	135 (90.0)	15 (10.0)	Reference category			
Yes	46 (88.4)	4 (11.6)	1.173	0.43	3.20	0.754
**Histological PTC variant**						
Classic PTC	125 (90.6)	13 (9.4)	NPC			
Encapsulated PTC variant	7 (100.0)	0 (0.0)	NPC			
Follicular PTC variant	18 (81.8)	4 (18.2)	NPC			
Aggressive PTC variants	31 (88.6)	4 (11.4)	NPC			0.641 ^&^
**Lymphocytic thyroiditis**						
Absent	146 (89.6)	17 (10.4)	Reference category			
Present	35 (89.7)	4 (10.3)	1.018	0.32	3.21	0.975

Values are n (%). CI: confidence interval. cOR: Crude odd ratio. uTg: undetectable thyroglobulin. dTg: detectable thyroglobulin. PTC: papillary thyroid cancer. NPC: no possible calculation of cOR and CI due to small number of patients in dTg group. &: Fisher’s exact test. Missing data: multicentricity 0 percent, histological PTC variant 0 percent, and lymphocytic thyroiditis 0 percent.

**Table 4 cancers-16-04129-t004:** Staging features of 202 patients undergoing total thyroidectomy due to papillary thyroid cancer, 2015–2023. Bivariate analysis among patients with undetectable or detectable basal postoperative thyroglobulin.

				95% CI for cOR		
	uTg 181 (89.6)	dTg 21 (10.4)	cOR	Inferior	Superior	*p* Value
**T stage**						
T1a + T1b	135 (93.1)	10 (6.9)	Reference category			
T2	26 (89.6)	3 (10.4)	1.557	0.40	6.04	0.522
T3a	10 (83.3)	2 (16.7)	2.7	0.51	14.03	0.238
T3b	9 (69.2)	4 (30.8)	6	1.56	22.95	0.009
T4	1 (33.3)	2 (66.7)	27	2.24	324.00	0.009
**N stage**						
N0 or Nx	135 (92.5)	11 (7.5)	Reference category			
N1a	26 (92.9)	2 (7.1)	0.990	0.12	7.68	0.764
N1b	20 (71.4)	8 (28.6)	5.156	0.96	27.44	0.118
**M stage**						
M0 or Mx	177 (92.2)	15 (7.8)	Reference category			
M1	4 (40.0)	6 (60.0)	17.7	4.49	69.69	0.000
**MACIS score**						
<6 points	1145 (95.8)	5 (4.2)	Reference category			
6–8 points	60 (89.5)	7 (10.5)	2.66	0.80	8.73	0.107
>8 points	7 (43.7)	9 (56.2)	29.314	7.72	111.20	0.000

Values are n (%). CI: confidence interval. cOR: crude odd ratio. uTg: undetectable thyroglobulin. dTg: detectable thyroglobulin. MACIS score: metastasis, age, completeness of resection, invasion, and size. Missing data: T stage 0 percent, N stage 0 percent, M stage 0 percent, and MACIS score 0 percent.

## Data Availability

The data presented in this study are available upon reasonable request to the corresponding author.

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
