# Peer review of "Predictive Factors of Athyroglobulinemia After Total Thyroidectomy for Papillary Thyroid Cancer"

_cancers, 2024, doi:10.3390/cancers16244129_

Round 1

Reviewer 1 Report

Comments and Suggestions for Authors

From a biostats and clinical epidemiology point of view, here are some comments for the Authors

- even in an observational retrospective single-center context, this study has been well planned, executed and reported; moreover, the list of selected covariates is clinically and epidemiologically consistent

- table 1 report either descriptive or inferential results, better to split it in 2 different ones

- the median follow-up from surgery to last contact for the entire cohort is lacking, it would be a super important clinical info

- a more immediate comment about the role of cOR would help the non-biostats reader! what message conveys to us!?

- missing data, this is an extremely good info! congrats!

- table 3 deserves to be splitted in 2 ones, as stated before

- the lackness of multivariable data, due to the reduced number of events, is probably the most serious concern, even we just know it

Author Response

Thank you very much for taking the time to review this manuscript. Please find the detailed responses below and the corresponding revisions and corrections highlighted in black and underlined in the re-submitted files.

Comment 1: even in an observational retrospective single-center context, this study has been well planned, executed and reported; moreover, the list of selected covariates is clinically and epidemiologically consistent

Response 1: Thank you for this comment, we truly appreciate it.

Comment 2: table 1 report either descriptive or inferential results, better to split it in 2 different ones.

Response 2: the table is now splitted in two different ones.

Comment 3: the median follow-up from surgery to last contact for the entire cohort is lacking, it would be a super important clinical info

Response 3: this information have been added in results, lines 273-274.

Comment 4: a more immediate comment about the role of cOR would help the non-biostats reader! what message conveys to us!?

Response 4: this is now explained in line 213.

Comment 5: missing data, this is an extremely good info! congrats!

Response 5: Thank you so much for this comment!

Comment 6: table 3 deserves to be splitted in 2 ones, as stated before

Response 6: this table is now splitted in two, like the previous table.

Comment 7: the lackness of multivariable data, due to the reduced number of events, is probably the most serious concern, even we just know it.

Response 7: We have included this limitation as the main one in the article for this reason. We believe it would be beneficial to conduct multicentric studies to avoid a small sample size of patients with dTg.

Reviewer 2 Report

Comments and Suggestions for Authors

The current manuscript investigates predictive factors for achieving undetectable thyroglobulin (uTg) levels after total thyroidectomy in patients with papillary thyroid cancer (PTC). This retrospective study analyzes 202 cases, identifying that uTg rates, a marker of effective surgical treatment, correlate more closely with tumor stage factors—such as tumor size, lymph node involvement, and metastatic spread—than with patient demographics or clinical symptoms. Extended resections and lymphadenectomy were linked to higher risks of detectable thyroglobulin, indicating more advanced disease. The findings highlight that successful biochemical outcomes after surgery depend on tumor-related factors, emphasizing the importance of early detection and comprehensive surgical strategies in PTC management.

1. It is not clear what is novel about this study.

2. Retrospective studies still require the prior approval of an ethical committee and be based on a prior written informed consent of the patient.

3. The subtitle "analyzed variables" in the methods is disorganized. The analyzed variables should be rearranged under subtitles and by using bullets, numbers or letters to list them.

4. In the results, line 213, uTg should be replaced by dTg.

5. Strengths and limitations of the study should be placed at the end of the discussion rather than its beginning.

6. The small number size should be listed as the major limitation of the current study as it greatly influenced the statistical power and the validity of the findings.

7. The languages of the manuscript should be revised.

Naturally, the presence of metastatic spread was a strong predictor of postoperative uTg.

Author Response

Thank you very much for taking the time to review this manuscript. Please find the detailed responses below and the corresponding revisions and corrections highlighted in black and underlined in the re-submitted files.

Comment 1: It is not clear what is novel about this study.

Response 1: This is now explained in the discussion, lines 417-421.

Comment 2: Retrospective studies still require the prior approval of an ethical committee and be based on a prior written informed consent of the patient.

Response 2: Also added at the end or the article, lines 436-442.

Comment 3: The subtitle "analyzed variables" in the methods is disorganized. The analyzed variables should be rearranged under subtitles and by using bullets, numbers or letters to list them.

Response 3: We have now rearranged under subtitles with numbers to list them.

Comment 4: In the results, line 213, uTg should be replaced by dTg.

Response 4: We have changed the entire sentence to ensure a better understanding of it, now line 218.

Comment 5: Strengths and limitations of the study should be placed at the end of the discussion rather than its beginning.

Response 5: We have changed it, now in lines 406-416.

Comment 6: The small number size should be listed as the major limitation of the current study as it greatly influenced the statistical power and the validity of the findings.

Response 6: We have listed this limitation as the primary one in the limitations section of the article, line 406.

Comment 7: The languages of the manuscript should be revised.

Response 7: We have reviewed the entire article and made some changes to the phrasing to ensure better understanding. These changes are highlited in black and italic.

Round 2

Reviewer 2 Report

Comments and Suggestions for Authors

The manuscript has been improved, however, the novelty of this work is still insufficient.

Additionally, I wonder what is the " local Research Ethics Committee of Granada (Spain)?" Is this an institutional ethical committee? What is the code of this approval? 

Author Response

Thank you very much for taking the time to review this manuscript again. Please find the detailed responses below and the corresponding revisions and corrections highlighted in black in the re-submitted files.

Comment 1: The manuscript has been improved, however, the novelty of this work is still insufficient.

Response 1: I have added some extra information about the novelty of this article.

Comment 2: Additionally, I wonder what is the " local Research Ethics Committee of Granada (Spain)?" Is this an institutional ethical committee? What is the code of this approval? 

Response 2: The Local Research Ethics Committee of Granada is an institutional ethics committee that is part of the network of ethics committees within the public healthcare system of Andalusia, Spain. Regarding the approval of this Committee, the information is provided in the document I attached to Mr. Asher. I will also submit it in the revised manuscript.
